# A two-gene marker for the two-tiered innate immune response in COVID-19 patients

Hongxing Lei[1,2,3]*

**1** CAS Key Laboratory of Genome Sciences and Information, Beijing Institute of Genomics, Chinese Academy of Sciences, and China National Center for Bioinformation, Beijing, China, **2** Cunji Medical School, University of Chinese Academy of Sciences, Beijing, China, **3** Center of Alzheimer's Disease, Beijing Institute for Brain Disorders, Beijing, China

* leihx@big.ac.cn

## Abstract

For coronavirus disease 2019 (COVID-19), a pandemic disease characterized by strong immune dysregulation in severe patients, convenient and efficient monitoring of the host immune response is critical. Human hosts respond to viral and bacterial infections in different ways, the former is characterized by the activation of interferon stimulated genes (ISGs) such as *IFI27*, while the latter is characterized by the activation of anti-bacterial associated genes (ABGs) such as *S100A12*. This two-tiered innate immune response has not been examined in COVID-19. In this study, the activation patterns of this two-tiered innate immune response represented by *IFI27* and *S100A12* were explored based on 1421 samples from 17 transcriptome datasets derived from the blood of COVID-19 patients and relevant controls. It was found that *IFI27* activation occurred in most of the symptomatic patients and displayed no correlation with disease severity, while *S100A12* activation was more restricted to patients under severe and critical conditions with a stepwise activation pattern. In addition, most of the *S100A12* activation was accompanied by *IFI27* activation. Furthermore, the activation of *IFI27* was most pronounced within the first week of symptom onset, but generally waned after 2–3 weeks. On the other hand, the activation of *S100A12* displayed no apparent correlation with disease duration and could last for several months in certain patients. These features of the two-tiered innate immune response can further our understanding on the disease mechanism of COVID-19 and may have implications to the clinical triage. Development of a convenient two-gene protocol for the routine serial monitoring of this two-tiered immune response will be a valuable addition to the existing laboratory tests.

## Introduction

The COVID-19 pandemic has relentlessly taken over 6.6 million lives in the past three years (https://covid19.who.int/). Although vaccination and drug treatment have saved millions of lives [1, 2], we have not seen the ending sight of the pandemic yet. The pandemic has presented major challenges to the global medical systems, including the overwhelmingly large volume of

**Data Availability Statement:** All of the datasets analyzed in the current study are publicly available. Most of the transcriptome datasets are available at GEO (gene expression omnibus, https://www.ncbi.nlm.nih.gov/geo/). The accession numbers are:

GSE152641, GSE171110, GSE167000, GSE172114, GSE189990, GSE161777, GSE161731, GSE155454, GSE166190, GSE178967, GSE169687, GSE154998, GSE157103, GSE152418, GSE184401, GSE179627 and GSE68310. One of the datasets is available at the Zenodo repository (https://zenodo.org/). The accession number is 6120249.

**Funding:** This work was supported by grants from the Strategic Priority Research Program of Chinese Academy of Sciences (Grant No. XDB38030200), the National Key Research and Development Program of China (grant no. 2016YFC0901700), and the National High Technology Program of China (863 Program; Grant No. 2015AA020108) awarded to HL by the Ministry of Science and Technology of China. The funders had no role in study design, data collection and analysis, decision to publish, or preparation of the manuscript.

**Competing interests:** HL has filed a patent application related to this work (Hongxing Lei, 202110367243.9). This does not alter our adherence to PLOS ONE policies on sharing data and materials."

hospitalized patients and fast deteriorating conditions for many patients. To meet these challenges, convenient and effective ways for continuous patient monitoring need to be deployed for quick and better clinical decision making.

To achieve this goal, a great variety of clinical tests have been explored in hospitals around the globe [3, 4]. Conventional blood tests included blood cell counts (especially neutrophils and lymphocytes) and size distribution [5–8]. Common laboratory tests included procalcitonin (PCT) [9], C-reactive protein (CRP) [10], D-dimer [11], interleukin-6 (IL-6) [12], and others [13]. In addition, parameters reflecting organ dysfunction have been extensively studied [14–17]. To assist diagnosis and prognosis, simple models have been constructed based on some of these parameters and demographic information such as age, sex, and number of comorbidities [18–22]. However, these models are far from perfect, reflecting the heterogeneity and complexity of COVID-19.

For more in-depth understanding of this disease, modern technologies have also been applied to the investigation. Deep immune profiling based on single-cell technologies have revealed profound change in the adaptive immunity [23–26]. Clinically relevant novel factors have been discovered using proteomics [27, 28], metabolomics [29, 30] and lipidomics [31, 32]. Mechanistic and biomarker studied have also employed transcriptomics of mRNA [33], miRNA [34], and cell-free RNA [35]. Unfortunately, the novel factors from these discovery investigations have rarely been replicated at independent institutions. Ideally, for the purpose of daily monitoring, the biomarkers need to be easily accessible and robustly reproducible under a variety of circumstances.

Gene activation or suppression in peripheral blood can be easily accessed and evaluated. In fact, activation of interferon stimulated genes is the most prominent signature of host response to viral infection [36], while activation of antibacterial associated genes has been consistently observed in patients with bacterial infection [37]. These two groups of genes represent a two-tiered innate immune response to infection and potentially other medical conditions. In previous works, we have proposed *S100A12* as the most prominent host marker for bacterial infection which was compared favorably against PCT and CRP in over a thousand clinical samples [38]. Others have extensively utilized *IFI27* and other ISGs for the identification of viral infection in clinics [39–41]. Since these two genes are directly linked to disease mechanism, the simultaneous monitoring of *IFI27* and *S100A12* in COVID-19 will give us a clearer insight into the two-tiered innate immune response to this devastating medical condition.

The goal of this work is to find out whether *IFI27* and *S100A12* can be used as effective markers to monitor COVID-19 progression. The following features will be examined: How prevalent is the gene activation in the blood of COVID-19 patients? Is the gene activation correlated with disease severity? Is the gene activation correlated with disease duration? Additionally, is the activation pattern consistent in whole blood and peripheral blood mononuclear cells (PBMCs)? The robustness of the patterns will be corroborated by the consistency among independent studies from around the globe. Following the investigation, it will be clear that the two genes can be used as valuable markers for the convenient and efficient monitoring of the pandemic disease.

## Materials and methods

### Transcriptome datasets for COVID-19

Most of the transcriptome datasets were downloaded from gene expression omnibus (GEO, https://www.ncbi.nlm.nih.gov/geo/). Only one of the datasets were downloaded from the Zenodo repository (https://zenodo.org/). All of these transcriptome studies on COVID-19

employed RNA-Seq (RNA sequencing) technology, while the only transcriptome study on seasonal flu utilized microarray technology.

The tissue source was whole blood for 12 of the 17 transcriptome datasets on COVID-19. Dataset **GSE152641** contains 86 samples, including 24 samples from healthy donors and 62 samples from COVID-19 patients [42]. Dataset **GSE171110** contains 54 samples, including 10 samples from healthy donors and 44 samples from severe COVID-19 patients [43]. Dataset **GSE167000** contains 95 samples from hospitalized subjects, including 30 samples from SARS-CoV-2 (severe acute respiratory syndrome coronavirus 2) negative patients and 65 samples from SARS-CoV-2 positive patients [44]. Dataset **GSE172114** contains 69 samples from COVID-19 patients, including 46 samples from critical patients and 23 samples from non-critical patients [45]. Dataset **Zenodo 6120249** contains 143 samples, including 10 samples from healthy volunteers, 13 samples from non-hospitalized COVID-19 patients, 78 samples from hospitalized COVID-19 patients, and 42 samples from sepsis patients [46]. Dataset **GSE189990** contains 24 samples, including 4 samples from healthy controls and 20 samples from COVID-19 patients [47]. Dataset **GSE161777** contains 56 samples, including 14 samples from healthy controls and 42 samples from COVID-19 patients [48]. Dataset **GSE161731** contains 155 samples, including 19 samples from healthy controls, 12 samples from hospitalized COVID-19 patients, 35 samples from non-hospitalized COVID-19 patients, 23 samples for patients with bacterial infection, 17 samples from patients with influenza infection, and 49 samples from patients with other coronavirus infection [49]. Dataset **GSE155454** contains 58 samples, including 6 samples from healthy controls and 52 samples from COVID-19 patients (reference not available). Dataset **GSE166190** contains 98 samples, including 35 samples from children and 63 samples from adults [50]. Dataset **GSE178967** contains 144 samples from COVID-19 patients at day 0 and day 5 of the hospital admission [51]. Dataset **GSE169687** contains 152 samples, including 14 samples from healthy controls and 138 samples from recovered COVID-19 patients at 3 time points [52].

The tissue source was leukocytes for two transcriptome datasets. Dataset **GSE154998** contains 14 samples from intensive care unit (ICU) patients, including 7 samples from SARS-CoV-2 negative patients and 7 samples from SARS-CoV-2 positive patients [53]. Dataset **GSE157103** contains 126 samples, including 100 samples from COVID-19 patients (50 ICU patients and 50 non-ICU patients) and 26 samples from patients with other diseases [54].

The tissue source was PBMCs for three transcriptome datasets. Dataset **GSE152418** contains 34 samples, including 17 samples from healthy controls and 17 samples from COVID-19 patients [55]. Dataset **GSE184401** contains 43 samples, including 21 samples from patients with mild COVID-19 and 22 samples from patients with severe COVID-19 (reference not available). Dataset **GSE179627** contains 70 samples, including 22 samples from healthy controls and 48 samples from COVID-19 patients [56].

All of the COVID-19 patients were tested positive for SARS-CoV-2 infection and had symptoms compatible with COVID-19 infection. In total, 1421 samples were included in these 17 RNA-Seq studies on COVID-19, including 1134 whole blood samples, 140 leukocyte samples, and 147 PBMC samples.

Additionally, dataset **GSE68310** contains 281 samples from non-hospitalized patients with mild influenza infection sampled day 2, day 4, day 6 and day 21 from symptom onset [57]. The tissue source was whole blood for this dataset.

## Data analysis and presentation

For the RNA-Seq datasets, expression values of *IFI27* and *S100A12* were extracted from the original expression matrix file and normalized based on the total number of reads for each

sample and then log transformed. The normalization step was skipped if the normalized data was already provided by the original data submitters. For the microarray dataset, the normalized data were downloaded from GEO and the expression values of *IFI27* and *S100A12* were extracted from the gene expression matrix. The gene expression and sample information were merged for further analysis. Empirical cut-off values for gene activation were usually based on the highest values in the healthy control groups. The correlation analysis was done in Excel. The group comparison (p-value calculation) for each dataset was done in R (https://www.r-project.org/) using t-test. The figures were drawn with ggplot2 package in R.

## Results

### Activation of *IFI27* and *S100A12* in COVID-19 patients

As host markers for infection, activation of both *IFI27* and *S100A12* was observed in a significant portion of patients with COVID-19 compared to healthy controls. In the dataset GSE152641 (Fig 1A), sampling was conducted within 24 hours of hospital admission with a median of 6 days from symptom onset (4–8 days). In this dataset, *S100A12* expression was significantly higher in the patient group (p = 1.019e-07). It was below 11.20 in all of the 24 healthy controls (median value 9.66), while it was above 11.20 in 25 of the 62 patients (40.3%) with COVID-19 (median value 11.03). In the meantime, *IFI27* expression was also significantly higher in the patient group (p = 1.09e-10). It was below 9.0 in 23 of the 24 healthy controls (median value 5.65, the other one being an outlier), while it was above 9.0 in 47 of these 62 patients (75.8%) with COVID-19 (median value 10.65). It showed that activation of both genes can be observed in the first week of the disease course, and activation of *IFI27* was more common and at larger scale than that of *S100A12* in hospitalized patients with COVID-19. Furthermore, activation of both genes was observed in 22 of the 62 patients (35.5%), suggesting that activation of *S100A12* was generally accompanied by the activation of *IFI27* (22 out of 25, 88%). In addition, non-activation of these two genes was observed in 12 of the 62 hospitalized patients (19.4%).

Since patients with severe disease are more concerned, the comparison between severe COVID-19 patients and healthy controls are of high interest. In the dataset GSE171110 (Fig 1B), 36 of the 44 severe patients were ICU patients, and sampling was conducted within 3 days of hospital admission with a median of 11 days from symptom onset (7–14 days). In this dataset, *S100A12* expression was significantly higher in the patient group (p = 1.076e-08). It was below 12.0 in all of the 10 healthy controls (median value 11.16), while it was above 12.0 in 35 of the 44 patients (79.5%) with severe COVID-19 (median value 13.33). In the meantime, *IFI27* expression was significantly higher in the patient group (p = 3.465e-05). It was below 11.0 in 9 of the 10 healthy controls (median value 7.57), while it was above 11.0 in 37 of these 44 patients (84.1%) with severed COVID-19 (median value 13.99). It showed that activation of these two genes were commonly observed in the second week of the disease course for severe patients. Compared to the dataset GSE152641, activation of *S100A12* was more common in patients with severe COVID-19. In addition, activation of both genes was observed in 29 of the 44 severe patients (65.9%), while non-activation of these two gene was only observed in one severe patient. Again, activation of *S100A12* was generally accompanied by the activation of *IFI27* (29 out of 35, 82.9%).

To have a broader understanding of gene activation, *IFI27* and *S100A12* expression can be compared between patients admitted to the same hospital with and without COVID-19. In the dataset GSE167000 (Fig 2A), all of the recruited non-ICU patients had relatively short hospital stay (2–9.5 days), some of which were tested positive for COVID-19. In this dataset, *S100A12* expression was similarly distributed in both groups (p = 0.402) with median values of 9.28 and

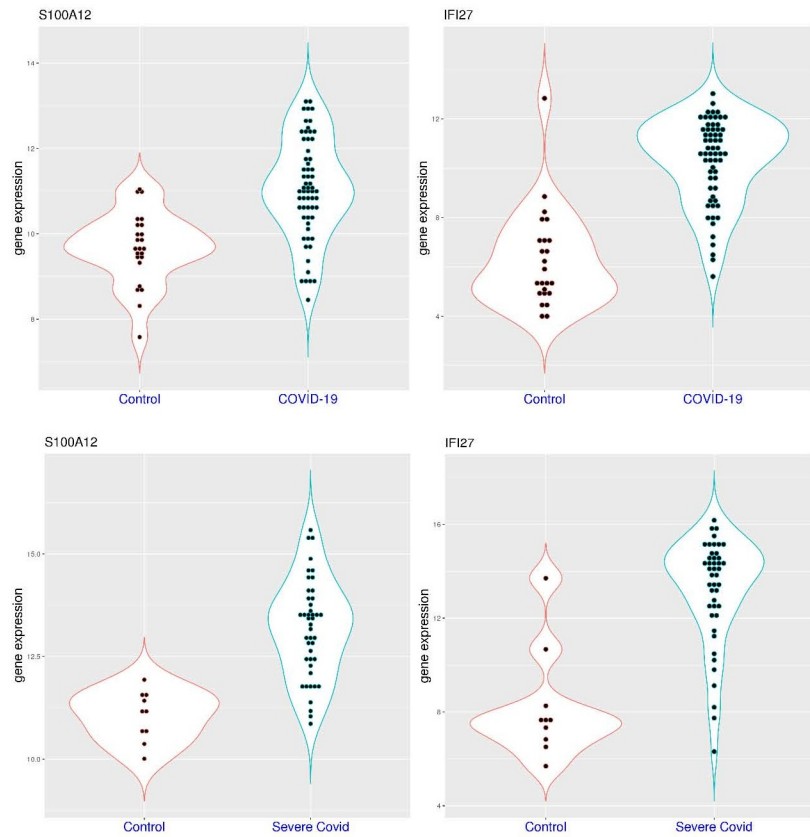

**Fig 1. Activation of *S100A12* and *IFI27* in a significant portion of hospitalized COVID-19 patients compared to healthy controls (whole blood). A)** upper panel (dataset GSE152641), comparison of *S100A12* and *IFI27* expression in healthy controls and hospitalized COVID-19 patients. **B)** lower panel (dataset GSE171110), comparison of *S100A12* and *IFI27* expression in healthy controls and severe COVID-19 patients.

9.22 in the patient groups with and without COVID-19, respectively. For example, *S100A12* expression was above 11.5 in only 1 of the 30 negative patients and 2 of the 65 positive patients. Interestingly, the highest value (13.8) was observed in the negative group, and the highest value in the positive group was below 12. In the meantime, *IFI27* expression was significantly higher in the positive group (median value 8.32) compared to the negative group (median value 2.74) (p = 6.413e-08). For example, *IFI27* expression was above 6.5 in 4 of the 30 negative patients (13.3%) and 45 of the 65 positive patients (69.2%). The top six *IFI27* expression values were all within the positive group. Overall, among hospitalized non-ICU patients, *S100A12* expression might not be distinguishable between the positive and negative groups, while *IFI27* activation was more restricted to the positive group.

The situation is somewhat different for ICU patients. In the dataset GSE154998 (Fig 2B), sampling was conducted at ICU admission for all of the recruited patients, some of which were tested positive for COVID-19. In this dataset, *S100A12* expression was mostly higher in the negative group (median values 6.76) compared to the positive group (median value 5.57) (p-value not calculated due to small sample size). The top four *S100A12* expression values were all within the negative group. On the other hand, *IFI27* expression was significantly higher in the positive group (median value 6.21) compared to the negative group (median value 0, not detected in most samples due to technical reasons). The *IFI27* expression values in the positive

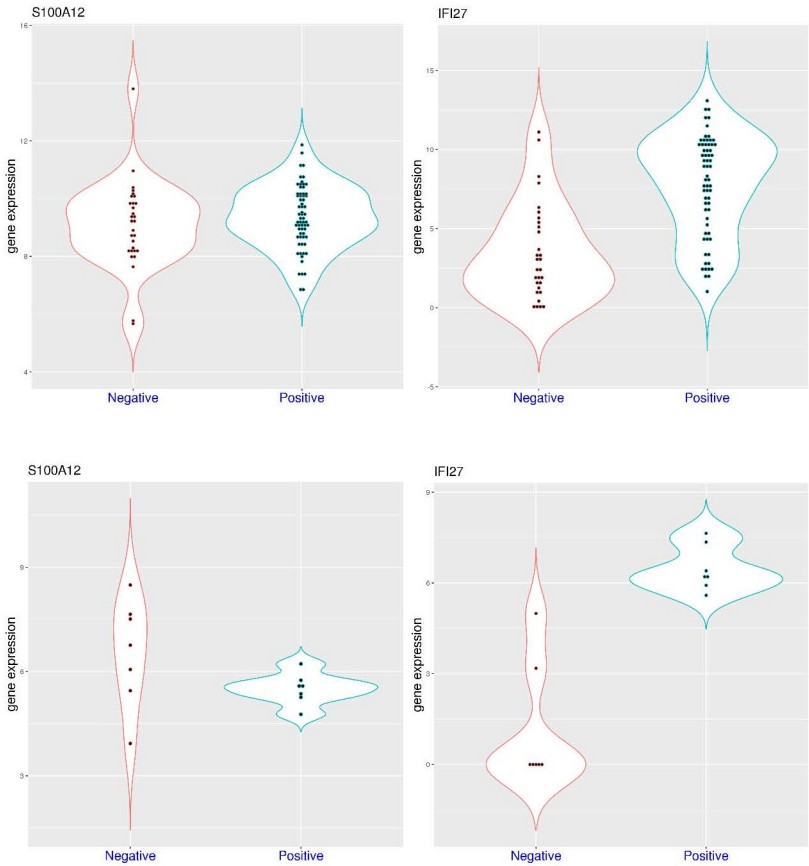

**Fig 2. Expression level of *S100A12* and *IFI27* in hospitalized patients with and without COVID-19 (whole blood).**
**A)** upper panel (dataset GSE167000), comparison of *S100A12* and *IFI27* expression in non-critical patients with and without COVID-19. **B)** lower panel (dataset GSE154998), comparison of *S100A12* and *IFI27* expression in intensive care unit (ICU) patients with and without COVID-19.

group were all higher than the top value in the negative group. Overall, among ICU patients, *S100A12* activation could be even stronger in certain patients within the negative groups (such as bacterial sepsis), while *IFI27* activation was more restricted to the positive group. Overall, for both ICU and non-ICU patients, *IFI27* activation was more specific to COVID-19.

## *S100A12* activation is correlated with COVID-19 severity

A practical question is whether the activation of *IFI27* or *S100A12* is correlated with COVID-19 severity. In the dataset GSE172114 (Fig 3A), sampling was conducted upon ICU admission (for critical patients, a median of 7 days post symptom onset) or hospital admission (for non-critical patients, a median of 9.5 days post symptom onset). In this dataset, *S100A12* expression was significantly higher in the critical patient group ($p = 1.285e\text{-}12$). It was above 6.0 in only 3 of the 23 patients (13.0%) with non-critical COVID-19 (median value 4.61), while it was above 6.0 in 41 of the 46 patients (89.1%) with critical COVID-19 (median value 7.63). In the meantime, *IFI27* expression was similarly distributed in both groups ($p>0.05$). It was above 5.0 in 15 of the 23 patients (65.2%) with non-critical COVID-19 (median value 6.46), while it was above 5.0 in 38 of the 46 patients (82.6%) with critical COVID-19 (median value 7.87). It showed that activation of *S100A12* was much more pronounced in critical COVID-19 patients

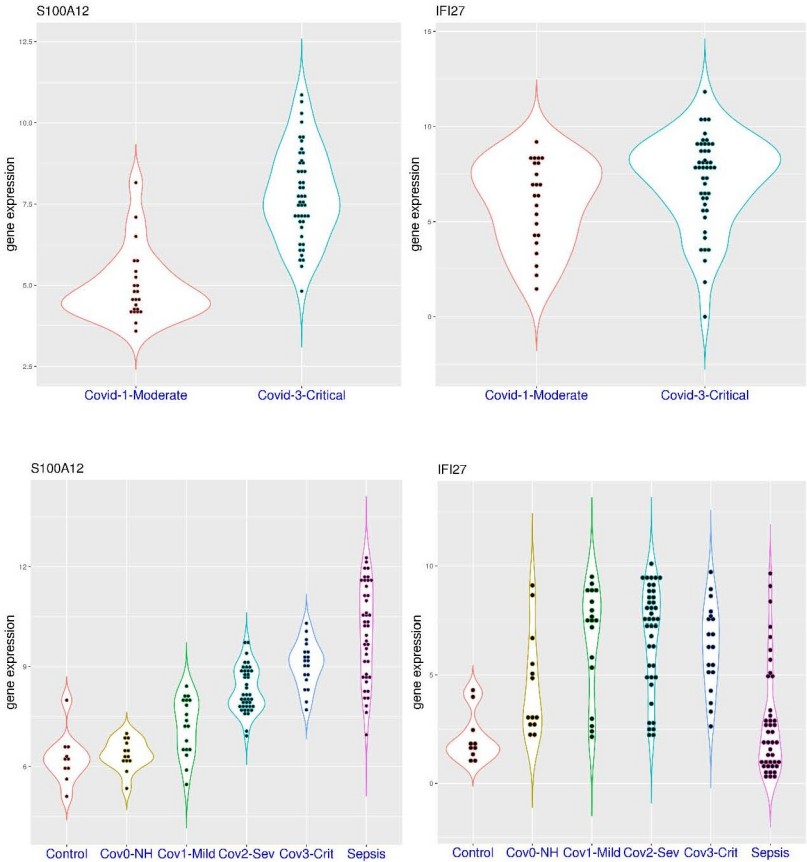

**Fig 3. Expression level of *S100A12* and *IFI27* in COVID-19 patients with different severity (whole blood). A)** upper panel (dataset GSE172114), comparison of *S100A12* and *IFI27* expression in moderate and critical COVID-19 patients. **B)** lower panel (dataset Zenodo 6120249), comparison of *S100A12* and *IFI27* expression in COVID-19 patients with different severity (non-hospitalized, moderate, severe and critical).

than that of non-critical counterparts. In addition, lower expression of both genes (empirical cutoff defined above) was not observed in patients with critical COVID-19, while higher expression of both genes was observed in 33 of those 46 patients (71.7%). Again, activation of *S100A12* was generally accompanied by the activation of *IFI27* (33 out of 41, 80.5%).

To have a strong correlation with disease severity, a step-wise increase of activation would be preferred. In the dataset Zenodo 6120249 (Fig 3B), sampling was conducted upon hospital or ICU admission (for hospitalized patients with varying time post symptom onset) or in the recovery stage (for community cases, at least 7 days post symptom onset). In this dataset, *S100A12* expression was significantly different in the COVID-19 patient groups with different severity. The p-values were 0.00142, 5.803e-05, and 0.000363 for the comparison of community cases to mild cases, mild cases to severe cases, and severe cases to critical cases, respectively. The expression was below 8.0 in all of the 10 healthy volunteers (median value 6.22) and same for all of the 13 non-hospitalized COVID-19 patients (median value 6.33), while it was above 8.0 in 4 of the 18 patients (22.2%) with mild COVID-19 (median value 7.31), 23 of the 41 patients (56.1%) with severe COVID-19 (median value 8.11), and 17 of the 19 patients (89.5%) with critical COVID-19 (median value 9.15). In addition, the highest value was only 8.41 in the mild COVID-19 group. Therefore, *S100A12* activation was a marker for severe and

critical COVID-19. In the meantime, *IFI27* expression was similar in the mild, severe and critical COVID-19 patient groups (p>0.05 for pairwise comparison). It was below 4.5 in all of the 10 healthy volunteers (median value 1.75), while it was above 4.5 in 6 of the 13 non-hospitalized COVID-19 patients (46.2%, median value 3.09), 14 of the 18 patients (77.8%) with mild COVID-19 (median value 7.57), 34 of the 41 patients (82.9%) with severe COVID-19 (median value 7.52), and 15 of the 19 patients (78.9%) with critical COVID-19 (median value 6.39). Overall, this clearly demonstrated a stepwise increase of *S100A12* activation in COVID-19, while *IFI27* activation was similarly predominant in hospitalized COVID-19 patients irrespective of disease severity. Again, activation of *S100A12* was generally accompanied by the activation of *IFI27* in COVID-19 patients (84.1% for this cohort). As an additional comparison, *S100A12* expression was above 8.0 in 39 of the 42 sepsis patients (92.9%, median value 10.23), while *IFI27* expression was above 4.5 in only 10 of these 42 sepsis patients (23.8%, median value 1.94). Therefore, activation of *S100A12* could be even stronger in sepsis, but it was infrequently accompanied by the activation of *IFI27* (except for viral sepsis).

This trend of step-wise *S100A12* activation in COVID-19 was replicated in two more datasets. In the dataset GSE189990 (Fig 4A), sampling was conducted upon hospital admission. In this dataset, *S100A12* expression showed a step-wise increase trend (p-values not calculated due to small sample size). It was below 8.0 in all of the 4 healthy controls (median value 7.27), while it was above 8.0 in the two patients (100%) with incidental COVID-19 (median value 8.31), all of the 4 patients (100%) with moderate COVID-19 (median value 9.44), 8 of the 9 patients (88.9%) with critical COVID-19 (median value 9.42), and all of the 5 patients (100%) with fatal COVID-19 (median value 11.16). In another dataset GSE161777 (Fig 4B), this trend could even be quantified by correlation analysis. In this dataset, multiple patients were sampled at multiple time points. The disease severity/stage was indexed as 2 to 7 with decreasing clinical score, representing critical, highly complicated, complicated, moderate, convalescence, and recovery. There was a moderate correlation (r = 0.54) between this severity index and *S100A12* expression.

This trend was also clear when combining the pictures from two more datasets. In the dataset GSE161731 (Fig 5A), hospitalized COVID-19 patients were compared with non-hospitalized patients and control subjects. In this dataset, *S100A12* expression was significantly higher in hospitalized COVID-19 patients compared to non-hospitalized patients (p = 0.000386). It was below 10.6 in all of the 19 healthy controls (median value 9.36), while it was above 10.6 in only one of the 35 non-hospitalized patients with COVID-19 (median value 8.72), and 5 of the 12 hospitalized patients (41.7%) with COVID-19 (median value 10.02). In the meantime, *IFI27* expression was below 6.2 in all of the 19 healthy controls (median value 1.0), while it was above 6.2 in 11 of the 35 non-hospitalized patients (31.4%) with COVID-19 (median value 4.3), and 11 of the 12 hospitalized patients (91.7%) with COVID-19 (median value 8.56). Overall, *S100A12* activation was rarely observed in non-hospitalized patients with COVID-19, while it could be observed in a significant portion of hospitalized patients with COVID-19, depending on the distribution of severity. For comparison, *S100A12* activation was not observed in the influenza group, rarely (1 out of 49) in the other coronavirus infection group (similar to the non-hospitalized COVID-19 group), but commonly (19 out of 23) in the bacterial infection group. On the other hand, *IFI27* activation was not observed in the bacterial infection group, infrequently (7 out of 49) in the other coronavirus infection group, but commonly (23 out of 23) in the influenza group.

In another dataset GSE157103 (Fig 5B), hospitalized patients were further divided into ICU patients and non-ICU patients. In this dataset, *S100A12* expression was significantly higher in the ICU group with COVID-19 compared to the non-ICU group (p = 7.815e-07). It was above 12.2 in 14 of the 50 non-ICU patients (28%) with COVID-19 (median value 11.74), while it

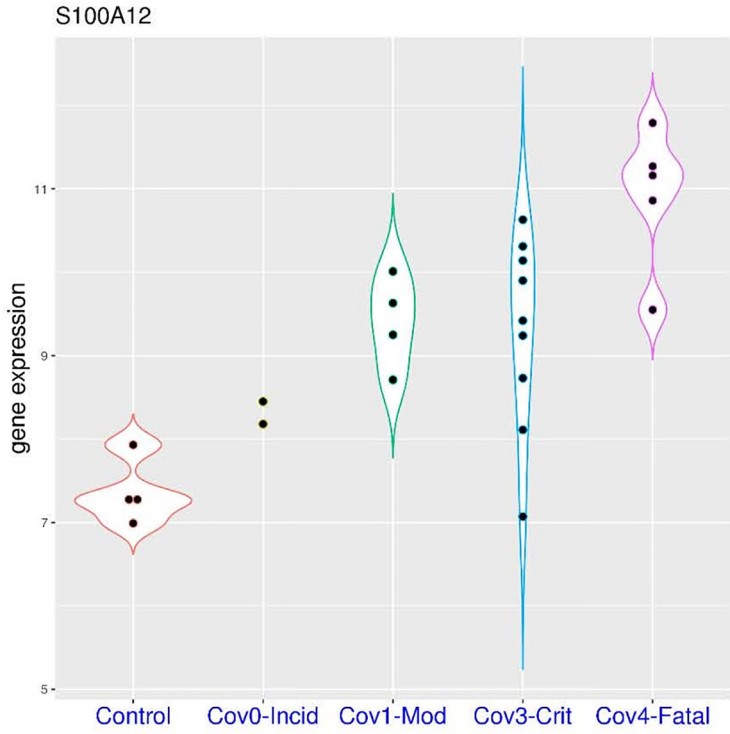

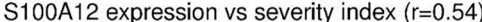

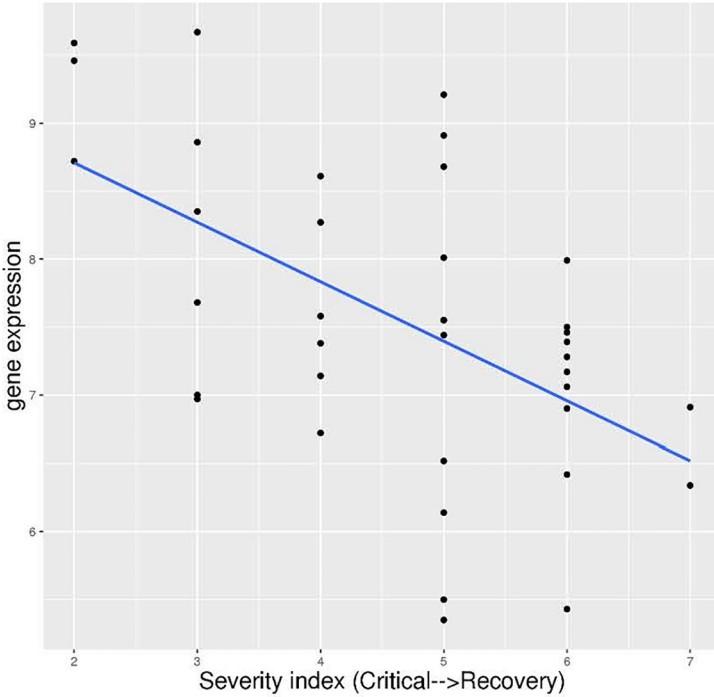

**Fig 4. Correlation of *S100A12* expression and the severity of COVID-19 (whole blood). A)** upper panel (dataset GSE189990), comparison of *S100A12* expression in COVID-19 patients with different severity (incidental, moderate, critical and fatal). **B)** lower panel (dataset GSE161777), correlation between *S100A12* expression and severity index (from critical to recovery).

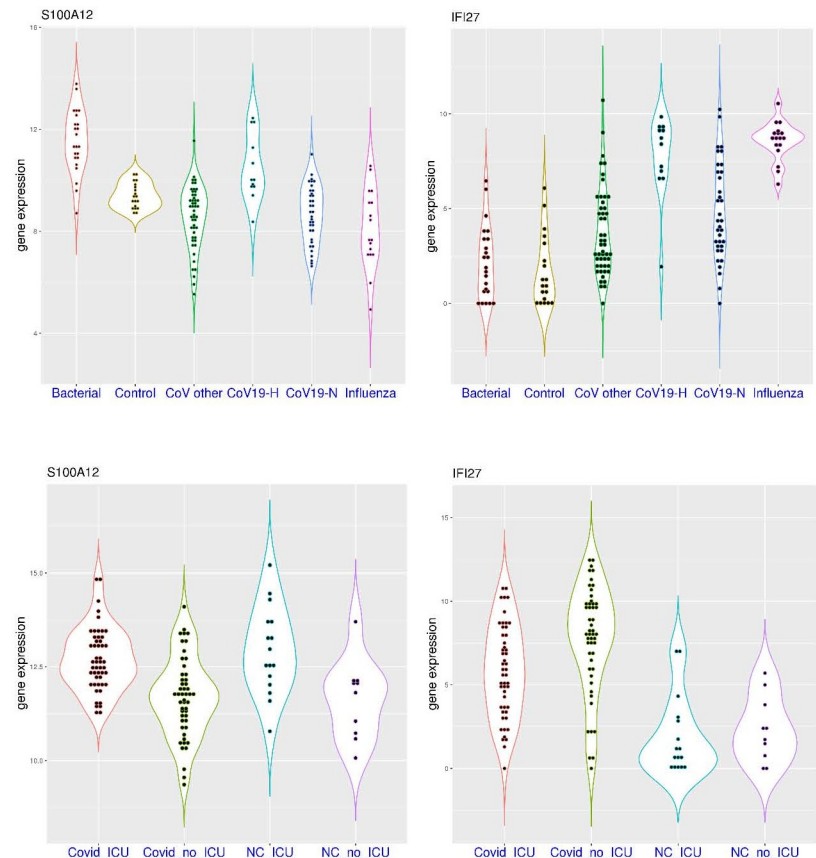

**Fig 5. Expression level of *S100A12* and *IFI27* in COVID-19 patients and patients with other diseases (whole blood). A)** upper panel (dataset GSE161731), comparison of *S100A12* and *IFI27* expression in healthy controls and patients with COVID-19 (hospitalized and non-hospitalized) or other types of infection. **B)** lower panel (dataset GSE157103), comparison of *S100A12* and *IFI27* expression in intensive care unit (ICU) and non-ICU patients with or without COVID-19.

was above 12.2 in 36 of the 50 ICU patients (72%) with COVID-19 (median value 12.62). In the meantime, *IFI27* expression was above 6.0 in 38 of the 50 non-ICU patients (76%) with COVID-19 (median value 8.11), and it was above 6.0 in 24 of the 50 ICU patients (48%) with COVID-19 (median value 5.74). In comparison, for patients without COVID-19, *S100A12* expression was above 12.2 in one of the 10 non-ICU patients, and 12 of the 16 ICU patients (75%), while *IFI27* expression was above 6.0 in none of the 10 non-ICU patients, and only 2 of the 16 ICU patients. Overall, *S100A12* activation was correlated with disease severity (with or without COVID-19), while *IFI27* activation was more specific to COVID-19. The combined observations from these two datasets (GSE161731 and GSE157103) showed the stepwise increase of *S100A12* activation in COVID-19 patients from non-hospitalized patients to hospitalized patients, and from hospitalized non-ICU patients to ICU patients.

### Fast waning of *IFI27* activation during COVID-19 disease course

As markers of host immune response, the dynamic properties of *IFI27* and *S100A12* expression can also be examined. In the dataset Zenodo 6120249 (Fig 6A), *IFI27* expression was moderately correlated with the time since symptom onset (r = 0.54). In another dataset GSE155454 (Fig 6B), *IFI27* expression displayed even stronger correlation with the time since symptom

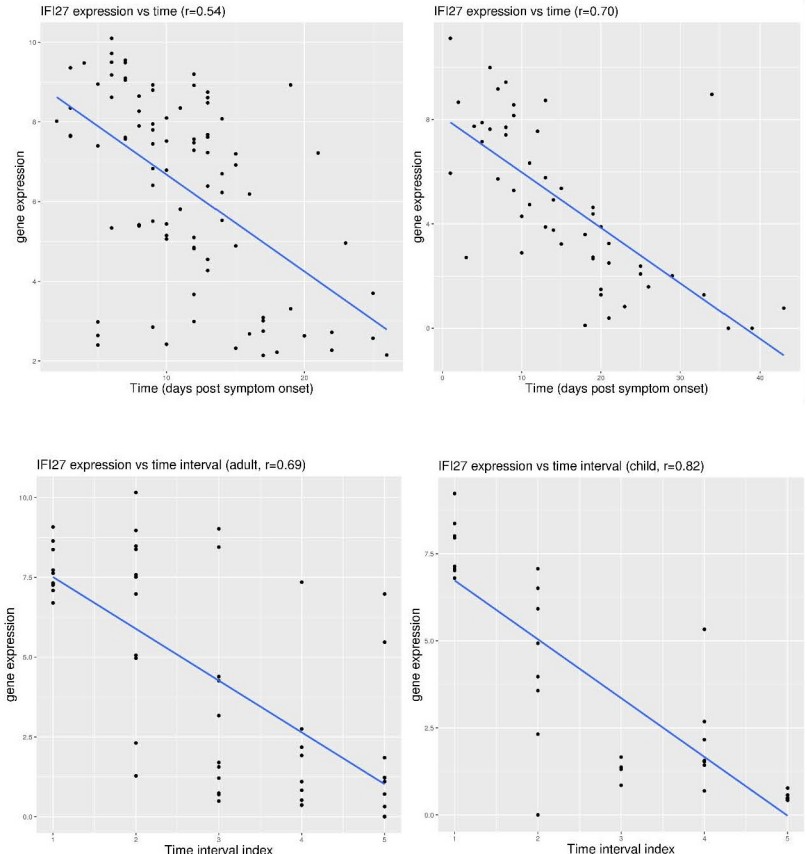

**Fig 6. Correlation of *IFI27* expression with the duration of COVID-19 infection. A)** upper left (dataset Zenodo 6120249), correlation between *IFI27* expression and time since symptom onset in COVID-19 patients. **B)** upper right (dataset GSE155454), correlation between *IFI27* expression and time since symptom onset in COVID-19 patients. **C)** lower panel (dataset GSE166190), correlation between *IFI27* expression and time interval index in COVID-19 patients (both adults and children).

onset (r = 0.70). In both datasets, *IFI27* expression was generally back to the control level after day 15. On the other hand, *S100A12* expression displayed no correlation with the time since symptom onset in both datasets (r close to 0).

This fast waning of *IFI27* activation was similarly observed in both adults and children. In the dataset GSE166190 (Fig 6C), time intervals were defined according to the days post symptom onset, including interval 1 (0–5 days), interval 2 (6–14 days), interval 3 (15–22 days), interval 4 (23–35 days), and interval 5 (36–81 days). In this dataset, *IFI27* expression displayed strong correlation with the time interval in both the adult group (r = 0.69) and the children group (r = 0.82). In both groups, *IFI27* expression was activated in the first time-interval (day 0–5), and mostly came down to the control level at time interval 3 (day 15 and after). This feature was consistent with the previous two datasets.

Not surprisingly, this fast waning of *IFI27* activation was also observed in seasonal influenza infection. In the dataset GSE68310 (Fig 7A), community monitoring of seasonal flu was conducted. Again, *IFI27* expression was already activated at day 2 after symptom onset and gradually came back to normal level before day 21. The correlation was also very strong between *IFI27* expression and the time since symptom onset (r = 0.87). Since the same group of people were sampled at multiple time points, this trend was even more convincing.

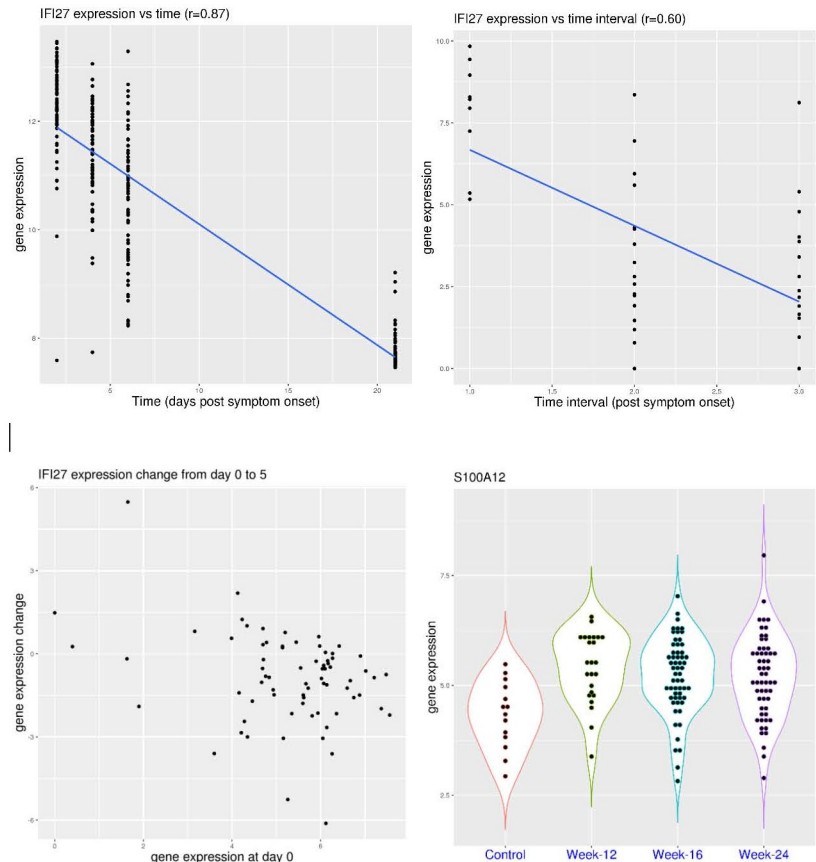

**Fig 7. Longitudinal tracking of the expression level of *IFI27* and *S100A12* during the course of infection. A)** upper left (dataset GSE68310), tracking of *IFI27* expression in a community cohort with seasonal influenza. **B)** upper right (dataset GSE161731), tracking of *IFI27* expression in non-hospitalized patients with COVID-19. **C)** lower left (dataset GSE178967), change of *IFI27* expression in COVID-19 patients 5 days after study enrollment. **D)** lower right (dataset GSE169687), long-term tracking of *S100A12* expression in COVID-19 patients.

Serial sampling was also conducted for patients with COVID-19. In the dataset GSE161731 (Fig 7B), some of the non-hospitalized COVID-19 patients were sampled at multiple time points and grouped into three time intervals (1-early, <11 days, 2-middle, 11–21 days, and 3-late, >21 days). Moderate correlation (r = 0.60) was observed between *IFI27* expression and the time interval. Activation of *IFI27* was observed in 7 of the 9 samples within time interval 1 (please refer to Fig 5A for the cutoff value), while only in 2 of 18 samples within time interval 2, and only in 1 of 16 samples within time interval 3. In another dataset GSE178967 (Fig 7C), COVID-19 outpatients were enrolled and examined at day 0 and 5 for gene expression (median days from symptom onset was 5 at enrollment). Compared to the expression level at the baseline, *IFI27* expression was down for most of the patients (52 out of 72, 72.2%) after only 5 days since enrollment. To be more specific, significant increase of *IFI27* expression (>1) only occurred to patients with low baseline expression (<4.5), while most of the patients (25 out of 28, 89.3%) with higher baseline expression (>6) displayed decrease of *IFI27* expression. Overall, the features from both datasets were consistent with the feature of fast waning of *IFI27* activation within the first two weeks.

The same does not apply to *S100A12* activation which may sustain for much longer time. In the dataset GSE169687 (Fig 7D), recovering COVID-19 patients were followed up to 6 months

from the first positive polymerase chain reaction (PCR) test. In this cohort, *S100A12* expression was below 6.0 in all of the 14 healthy controls (median value 4.41), but it was above 6.0 in 9 of the 24 patients at week 12 (median value 5.51), 12 of 58 patients at week 16 (median value 5.32), and 11 of the 56 patients at week 24 (median value 5.17). Although higher activation of *S100A12* (>7.0) was rarely observed (0 at week 12, and only one at week 16 and 24 each), this moderate activation in a significant portion of patients may still be related to the long-term health, since long COVID was observed later for some of these patients.

## Activation pattern of *IFI27* and *S100A12* in PBMC

The observations above were found in studies with whole blood (or leukocytes). A few other studies were conducted using PBMC. In the dataset GSE152418, sampling was mostly conducted within 3 weeks post symptom onset. In this dataset (Fig 8A), *S100A12* expression was below 11.6 in all of the 17 healthy volunteers (median value 10.01) and same for all of the 4 patients with moderate COVID-19 (median value 9.75), while it was above 11.6 in 4 of the 8 patients (50%) with severe COVID-19 (median value 11.62), and all of the 4 patients (100%) with critical COVID-19 (median value 12.96). This clearly demonstrated a step-wise increase of *S100A12* activation in the COVID-19 patients with increasing severity (p-value not calculated due to small sample size). In the meantime, *IFI27* expression was below 7.0 in all of the 17 healthy volunteers (median value 4.0), while it was above 7.0 in 15 of the 16 COVID-19 patients (median value 13.3, 10.33, and 9.98 for the moderate, severe and critical group, respectively). In addition (Fig 8B), *IFI27* expression was highly correlated with disease duration (r = 0.82), while *S100A12* expression displayed no correlation with disease duration (r = 0.1). Overall, the stepwise increase of *S100A12* activation and fast waning of *IFI27* activation were consistently observed in whole blood and PBMC of patients with COVID-19.

This feature was further replicated in two more datasets for PBMC. In the dataset GSE184401 (Fig 9A), severe patients were further divided into two groups, with or without secondary infection. In this dataset, *S100A12* expression was significantly higher in the severe patient group compared to the mild patient group (p = 8.989e-08). It was below 8.15 in all of the 21 patients with mild COVID-19 (median value 6.43), while it was above 8.15 in 11 of the 17 severe COVID-19 patients without secondary infection (median value 9.23), and all of the 5 severe COVID-19 patients with secondary infection (median value 8.72). On the other hand, *IFI27* expression was not significantly higher in the severe group compared to the mild group (p>0.05). The median value of *IFI27* expression was 3.51, 3.3, and 1.44 for the mild group, severe without secondary infection group, and severe with secondary infection group, respectively. For example, *IFI27* expression was above 5.0 in 6 of the 21 patients with mild COVID-19, 4 of the 17 severe COVID-19 patients without secondary infection, and one of the 5 severe COVID-19 patients with secondary infection.

In another dataset GSE179627 (Fig 9B), in addition to patients undergoing treatment (asymptomatic, mild or moderate), the study also included recovering patients at the time of hospital discharge and those who were tested positive again weeks after being discharged. In this dataset, *S100A12* expression was below 7.0 in all of the 22 control subjects, as well as all of the 21 hospitalized patients, while it was above 7.0 in 4 of the 15 recovering patients, 3 of the 12 re-positive patients. This showed that *S100A12* activation may last for a long time in certain COVID-19 patients. On the other hand, *IFI27* activation was mostly restricted to the acute phase of the disease. For example, *IFI27* expression was below 5.0 in all of the 22 control subjects as well as all of the 8 asymptomatic patients, while it was above 5.0 in 2 of the 3 mild patients, 7 of the 10 moderate patients, 2 of the 15 recovering patients, and none of the 12 re-

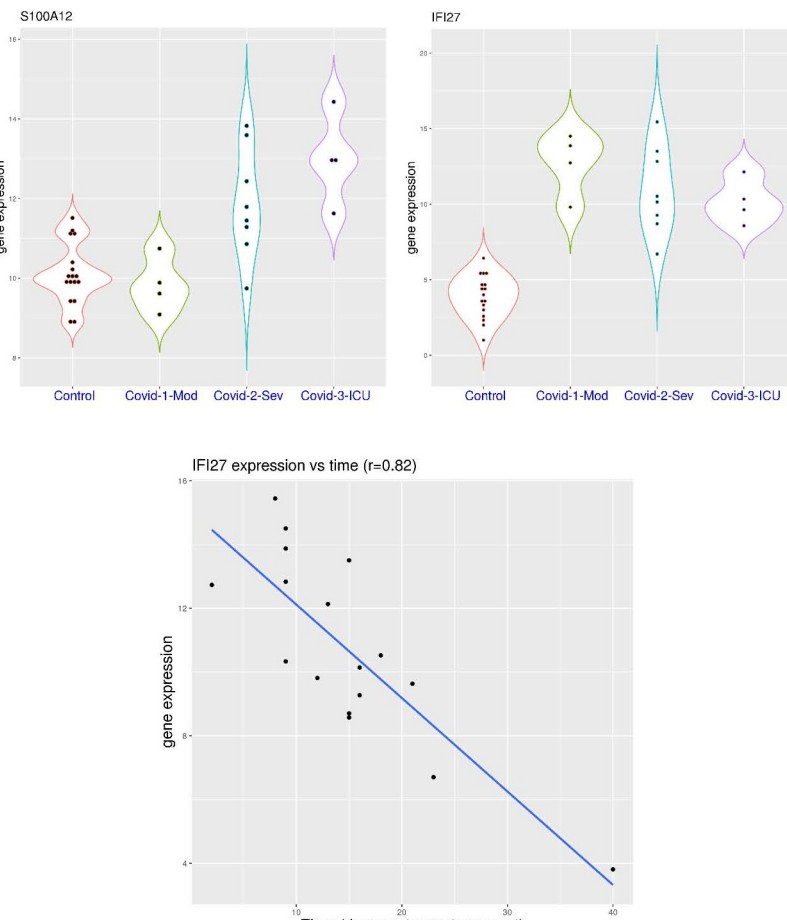

**Fig 8. Expression level of *S100A12* and *IFI27* in peripheral blood mononuclear cells (PBMCs) as correlated with COVID-19 severity or disease duration (dataset GSE152418). A)** upper panel, comparison of *S100A12* and *IFI27* expression in healthy controls and COVID-19 patients with different severity (moderate, severe and critical). **B)** lower panel, correlation between *IFI27* expression and time since symptom onset in COVID-19 patients.

positive patients. Overall, the fast waning of *IFI27* activation and the long tail of *S100A12* activation were consistently observed in whole blood and PBMC.

## Discussion

Based on the data presented above, a two-tiered innate immune response was clearly observed in patients with COVID-19, the first tier being the activation of ISGs, while the second tier being the activation of ABGs. Because interferon signaling is likely a universal response to respiratory viral infection, the activation of ISGs was highly prevalent in patients with symptomatic COVID-19. However, unlike seasonal flu, the activation of ABGs was observed in a significant portion of patients with COVID-19, along with much higher death rate in this pandemic disease [58]. This is likely because the first-tier response is far from sufficient for some patients with COVID-19, leading to the ensuing deployment of the second-tier response. The second-tier response is more restricted to severe or even critical patients, reflecting the overwhelming infection status including the high load of viral antigen and RNA in the blood [59]. Clearly, this response is not a mere reflection of neutrophil expansion [5], because similar pattern was observed in PBMCs.

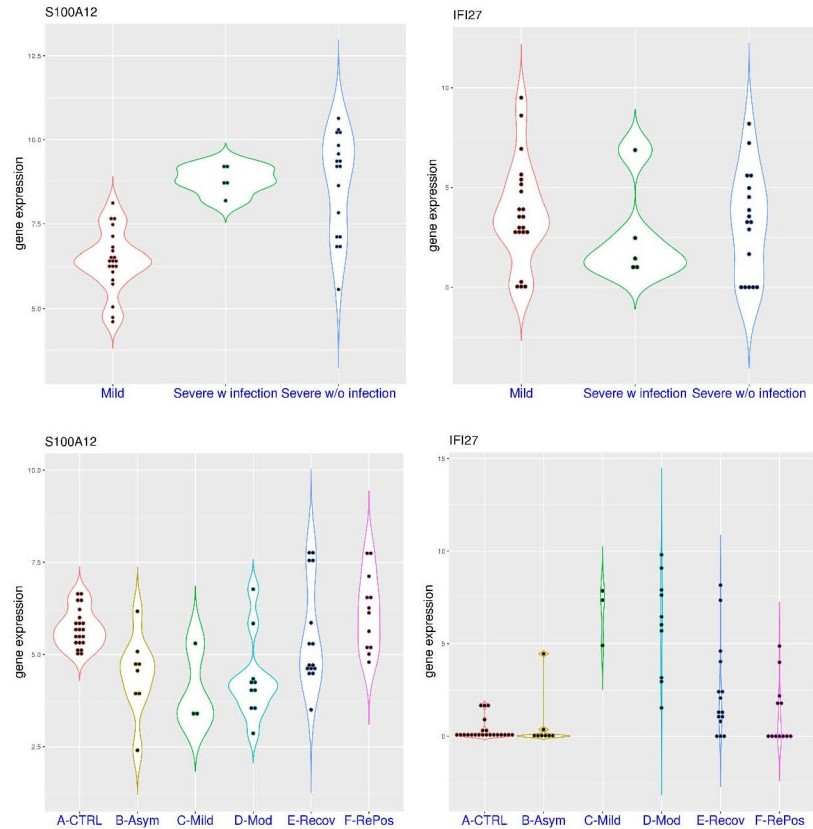

**Fig 9. Expression levels of *S100A12* and *IFI27* in peripheral blood mononuclear cells (PBMCs) as markers for the infection in COVID-19 patients. A)** upper panel (dataset GSE184401), comparison of *S100A12* and *IFI27* expression in COVID-19 patients with different severity (mild or severe) and secondary infection (with or without). **B)** lower panel (dataset GSE179627), comparison of *S100A12* and *IFI27* expression in healthy controls and COVID-19 patients with different severity (asymptomatic, mild and moderate) and different stages (acute, recovering and re-positive).

This study included 1421 samples from 17 independent studies around the globe, including Europe, North America, Asia and Australia. The consistent features from independent studies suggest that geographic locations do not have a significant impact on the two-tiered innate immune response in COVID-19 patients. Our previous works also demonstrated that age and sex do not affect the expression of the marker genes for the two-tiered innate immune response [38]. Thus, the observations presented above are robust and not confounded by these common factors.

An advantage of this two-gene marker panel is the direct link to the two-tiered innate immune response which makes it different from other markers or marker panels. Importantly, the first-tier response is strongly correlated with disease duration, while the second-tier response is strongly correlated with disease severity. Thus, rich clinically relevant information can be obtained through simultaneous monitoring of the two-tiered innate immune response. This has a clear advantage over existing markers such as PCT and CRP.

The characteristic features of this two-tiered response in COVID-19 can have implications in clinical decisions. Activation of several inflammatory markers has been observed in patients with severe COVID-19 and has been proposed as prognostic markers [19, 60, 61]. Since *S100A12* expression displayed a stepwise activation pattern and demonstrated strong association with severe COVID-19, it may serve as a good prognostic marker as well. In fact, *S100A12*

expression showed superior prognostic power compared to PCT, CRP, D-dimer and other parameters in our previous reanalysis of data from a prospective study [62]. This is likely because the second-tier innate immune response is highly connected to the disease mechanism in severe COVID-19 including hyperinflammation. Several studies showed that better prognostic power can be achieved by serial sampling of clinical parameters [63, 64]. Thus, serial sampling of *S100A12* expression is recommended, and sustained or even increased activation shall be taken as an alarming sign. Since *S100A12* activation may last for several months in certain patients with COVID-19, it warrants further investigation whether this sustained activation is associated with some of the lingering symptoms in long COVID [65]. In addition, another application is to rule out secondary bacterial co-infection when *S100A12* activation is not detected, similar to the recently proposed usage of PCT [66]. This is because *S100A12* activation is the most prominent signature of bacterial infection (with or without viral co-infection).

On the other hand, the fast waning of *IFI27* activation suggests that *IFI27* expression may serve as a surrogate of disease duration in patients with COVID-19. In the meantime, patients with sustained *IFI27* activation over 2–3 weeks may deserve special attention. Another application is to alert nosocomial SARS-CoV-2 infection for patients admitted for other reasons when *IFI27* activation is newly detected (compared to baseline value at admission), because *IFI27* activation is rarely observed in hospitalized patients with other diseases. Additionally, since most of the *S100A12* activation is accompanied by *IFI27* activation in patients with severe COVID-19, it is unlikely that the inability of ISG activation is to blame for most of the severe symptoms, which has implications in the usage of interferon as a therapeutic option.

This study is limited by the availability of more high-quality transcriptome data with enriched clinical information of individual patients, likely due to privacy concern and other practical reasons. For example, laboratory test results and time from symptom onset for each sample are not provided in most of the datasets. In addition, sample size is not ideal in many studies. It's also desirable to have high quality data with the inclusion of symptomatic patients, asymptomatic patients and healthy controls in the same study. Sampling at multiple time points is critical for disease monitoring but is rarely conducted in these transcriptome studies. In the case of long COVID, follow-up well over six months may be needed. It's also of strong interest whether and how the activation pattern may change with different variants of the virus.

This ongoing pandemic has made reverse transcriptase PCR (RT-PCR) platforms widely accessible and cost effective (~4$/test in mainland China), which can facilitate the realization of routine monitoring of the two-tiered innate immune response. In our previous works, we have conducted RT-PCR experiments on *S100A12*, *IFI27* and other relevant genes in well over a thousand clinical blood samples [38, 67, 68]. We have also designed a single-tube assay for three genes including *S100A12*, *IFI27* and an internal reference gene (unpublished work). Hopefully, this kind of assays can be applied to the routine monitoring of COVID-19 and other infectious diseases in the future.

## Conclusions

In this study, a two-tiered innate immune response was demonstrated in COVID-19 patients, and a two-gene marker (*IFI27/S100A12*) for this two-tiered innate immune response was proposed. For most of the symptomatic COVID-19 patients, *IFI27* activation was observable within the first 2–3 weeks of symptom onset and gradually came back to the baseline level after this acute phase. On the other hand, *S100A12* activation displayed no correlation with disease duration but was highly correlated with disease severity. Simultaneous serial monitoring of

these two genes can examine the two-tiered innate immune response in real time and may better guide clinical triage of COVID-19.

## Author Contributions

**Conceptualization:** Hongxing Lei.

**Data curation:** Hongxing Lei.

**Formal analysis:** Hongxing Lei.

**Funding acquisition:** Hongxing Lei.

**Investigation:** Hongxing Lei.

**Methodology:** Hongxing Lei.

**Project administration:** Hongxing Lei.

**Resources:** Hongxing Lei.

**Software:** Hongxing Lei.

**Supervision:** Hongxing Lei.

**Validation:** Hongxing Lei.

**Visualization:** Hongxing Lei.

**Writing – original draft:** Hongxing Lei.

**Writing – review & editing:** Hongxing Lei.

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
