## [Decision Letter · Decision Letter 0]

17 Nov 2022

PONE-D-22-29608Toward routine monitoring of the two-tiered innate immune response in COVID-19 patientsPLOS ONE

Dear Dr. Lei,

Thank you for submitting your manuscript to PLOS ONE. After careful consideration, we feel that it has merit but does not fully meet PLOS ONE’s publication criteria as it currently stands. Therefore, we invite you to submit a revised version of the manuscript that addresses the points raised during the review process.

We look forward to receiving your revised manuscript.

Kind regards,

Benjamin M. Liu, MBBS, PhD, D(ABMM), MB(ASCP)

Academic Editor

PLOS ONE

“This work was supported by grants from the Strategic Priority Research Program of Chinese Academy of Sciences (Grant No. XDB38030200), the National Key Research and Development Program of China (grant no. 2016YFC0901700), and the National High Technology Program of China (863 Program; Grant No. 2015AA020108) awarded to HL by the Ministry of Science and Technology of China. “

“This work was supported by grants from the Strategic Priority Research Program of Chinese Academy of Sciences (Grant No. XDB38030200), the National Key Research and Development Program of China (grant no. 2016YFC0901700), and the National High Technology Program of China (863 Program; Grant No. 2015AA020108) awarded to HL by the Ministry of Science and Technology of China.”

4. We note that you have a patent relating to material pertinent to this article. Please provide an amended statement of Competing Interests to declare this patent (with details including name and number), along with any other relevant declarations relating to employment, consultancy, patents, products in development or modified products etc. Please confirm that this does not alter your adherence to all PLOS ONE policies on sharing data and materials, as detailed online in our guide for authors http://journals.plos.org/plosone/s/competing-interests by including the following statement: "This does not alter our adherence to  PLOS ONE policies on sharing data and materials.” If there are restrictions on sharing of data and/or materials, please state these. Please note that we cannot proceed with consideration of your article until this information has been declared.

Reviewers' comments:

Reviewer's Responses to Questions

**Comments to the Author**

1. Is the manuscript technically sound, and do the data support the conclusions?

Reviewer #1: Yes

Reviewer #2: Partly

2. Has the statistical analysis been performed appropriately and rigorously? 

Reviewer #1: Yes

Reviewer #2: No

3. Have the authors made all data underlying the findings in their manuscript fully available?

Reviewer #1: Yes

Reviewer #2: No

4. Is the manuscript presented in an intelligible fashion and written in standard English?

Reviewer #1: Yes

Reviewer #2: Yes

5. Review Comments to the Author

Reviewer #1: Comments

1. The found the MS very fascinating and well organized.

2. The biomarkers you discussed “IFI27& S100A12” are novel regarding COVID-19 severity? but you only discussed it in symptomatic subjects is it possible? if you also take sample of asymptomatic individuals and compare the activation level of both population.

3. Rephrase line 85-92.

Reviewer #2: Comments to the author for evaluating the following manuscript

Toward routine monitoring of the two-tiered innate immune response in COVID-19 patients

The authors addressed the two-tiered innate immune response has not been examined in COVID-19.

It is greatly suggested that the manuscript is not ready to be accepted now. I have several comments listed below.

• The tile is not specified for the findings of the study. It must be rewritten to include also the criteria tested.

• The aim of the work in the abstract should be cleared.

• Introduction must be focus on the problem the research dealt with and how will the authors solve this problem in addition to determining the gap in this point in previous researches.

• Is the author depending on databases only for its study?

• Are these databases sufficient for analysis?

• Did the geographical distribution have a role in the data analysis?

• What is the impact of the patient’s age on the severity of the disease and the expression of two-tiered innate immune response?

• The discussion must include the causes for the similarities or differences for all items not only mention what are different or similar than other researches.

• What is the author interpretation if there is any mixed infection in the patients analyzed?

• The author must add a brief conclusion for his study.

• Where is the statistical analysis for this study?

Typo issues:

• There were some errors in the structure of several sentences.

• Under all figures, all abbreviations must be mentioned as footnotes.

• The authors must write what each abbreviated word stands for before using the abbreviation for the first time.

Finally, I cannot accept publishing of this article unless all corrections are made. Therefore, the manuscript is accepted for publication after major revisions

6. PLOS authors have the option to publish the peer review history of their article (what does this mean?). If published, this will include your full peer review and any attached files.

Reviewer #1: No

Reviewer #2: No

---

## [Author Response · Author response to Decision Letter 0]

20 Nov 2022

Reviewer #1: Comments

1. I found the MS very fascinating and well organized.

I appreciate the encouragement from the reviewer.

2. The biomarkers you discussed “IFI27& S100A12” are novel regarding COVID-19 severity. but you only discussed it in symptomatic subjects. is it possible if you also take sample of asymptomatic individuals and compare the activation level of both population.

To our knowledge, both the concept of two-tiered innate immune response and the two-gene marker panel are novel for COVID-19. As for the asymptomatic patients, the main aim of this study is to find convenient and efficient markers to monitor the innate immune response in hospitalized COVID-19 patients. Since asymptomatic patients rarely go to hospitals, they can not be monitored by medical professionals anyway. For the same reason, asymptomatic patients are not the focus of many studies. Nevertheless, asymptomatic or non-hospitalized patients were included in a few datasets analyzed in this study. But it’s uncertain how informative it is. Ideally, we would like to have reasonable sample size for asymptomatic and symptomatic patients as well as healthy controls in the same dataset. This statement has been added to the Discussion section.

3. Rephrase line 85-92.

The entire paragraph has been revised.

Reviewer #2: Comments to the author for evaluating the following manuscript

Toward routine monitoring of the two-tiered innate immune response in COVID-19 patients

The authors addressed the two-tiered innate immune response has not been examined in COVID-19.

It is greatly suggested that the manuscript is not ready to be accepted now. I have several comments listed below.

• The tile is not specified for the findings of the study. It must be rewritten to include also the criteria tested.

Thanks! The title has been rephrased.

• The aim of the work in the abstract should be cleared.

The aim of the work has been added to the beginning of the Abstract.

• Introduction must be focus on the problem the research dealt with and how will the authors solve this problem in addition to determining the gap in this point in previous researches.

Sorry, the aim of this study was not clearly stated in the previous version of Abstract. It has been revised. The Introduction section has also been revised accordingly.

• Is the author depending on databases only for its study?

Yes, this study is based on reanalysis of public datasets.

• Are these databases sufficient for analysis?

This study included 1421 samples from 17 independent datasets generated around the globe. The two-tiered innate immune response and the activation pattern of the two marker genes are well reproduced across so many independent datasets. Therefore, the results are quite robust.

• Did the geographical distribution have a role in the data analysis?

The samples were collected from around the globe, including Europe, North America, Asia and Australia. The patterns were consistent in different datasets.

• What is the impact of the patient’s age on the severity of the disease and the expression of two-tiered innate immune response?

Based on our previous works, age does not have an effect on the expression of IFI27 or S100A12. But it’s a well-known fact that age is a major contributing factor to COVID-19 severity. This statement has been added to the Discussion section in the revised manuscript.

• The discussion must include the causes for the similarities or differences for all items not only mention what are different or similar than other researches.

The Discussion section has been revised accordingly.

• What is the author interpretation if there is any mixed infection in the patients analyzed?

According to previous studies, bacterial co-infection in COVID-19 patients is not as common as expected earlier (<5%). Therefore, it’s unlikely to have a major effect on the analysis. A comparison of COVID-19 patients with or without coinfection is presented in Figure 9A.

• The author must add a brief conclusion for his study.

A Conclusion section has been added to the revised manuscript.

• Where is the statistical analysis for this study?

For datasets with sufficiently large sample, p-values for group comparison have been calculated and added to the revised manuscript.

Typo issues:

• There were some errors in the structure of several sentences.

The manuscript has been extensively revised to avoid errors.

• Under all figures, all abbreviations must be mentioned as footnotes.

This issue regarding figure captions has been taken care of.

• The authors must write what each abbreviated word stands for before using the abbreviation for the first time.

The issue regarding abbreviations has been taken care of.

---

## [Decision Letter · Decision Letter 1]

12 Dec 2022

PONE-D-22-29608R1A two-gene marker for the two-tiered innate immune response in COVID-19 patientsPLOS ONE

Dear Dr. Lei,

Thank you for submitting your manuscript to PLOS ONE. After careful consideration, we feel that it has merit but does not fully meet PLOS ONE’s publication criteria as it currently stands. Therefore, we invite you to submit a revised version of the manuscript that addresses the points raised during the review process.

We look forward to receiving your revised manuscript.

Kind regards,

Benjamin M. Liu, MBBS, PhD, D(ABMM), MB(ASCP)

Academic Editor

PLOS ONE

Journal Requirements:

Reviewers' comments:

Reviewer's Responses to Questions

**Comments to the Author**

1. If the authors have adequately addressed your comments raised in a previous round of review and you feel that this manuscript is now acceptable for publication, you may indicate that here to bypass the “Comments to the Author” section, enter your conflict of interest statement in the “Confidential to Editor” section, and submit your "Accept" recommendation.

Reviewer #1: All comments have been addressed

Reviewer #2: (No Response)

2. Is the manuscript technically sound, and do the data support the conclusions?

Reviewer #1: Yes

Reviewer #2: Yes

3. Has the statistical analysis been performed appropriately and rigorously? 

Reviewer #1: Yes

Reviewer #2: Yes

4. Have the authors made all data underlying the findings in their manuscript fully available?

Reviewer #1: Yes

Reviewer #2: Yes

5. Is the manuscript presented in an intelligible fashion and written in standard English?

Reviewer #1: Yes

Reviewer #2: Yes

6. Review Comments to the Author

Reviewer #1: Comments:

1. The author is suggested to revise the manuscript, and thoroughly read for English correction.

Reviewer #2: Comments for evaluating the following manuscript

A two-gene marker for the two-tiered 1 innate immune response in COVID-19 patients

The authors addressed the two-tiered innate immune response has not been examined in COVID-19.

I would like to thank the author for revision, but I have some comments listed below.

• The goal must be written at the end of the introduction section and the authors must arrange the ideas in introduction serially to introduce the problem and how to solve. Moreover, the introduction must not have detailed materials and methods.

• The author should illustrate the role of geographical distribution on his results.

• The discussion needs to be improved regarding the causes for the similarities or differences for majority of the items.

• Under all figures, there were many abbreviations those must be mentioned as footnotes.

7. PLOS authors have the option to publish the peer review history of their article (what does this mean?). If published, this will include your full peer review and any attached files.

Reviewer #1: No

Reviewer #2: No

---

## [Author Response · Author response to Decision Letter 1]

21 Dec 2022

Reviewer #1: Comments:

1. The author is suggested to revise the manuscript, and thoroughly read for English correction.

Thanks! The whole manuscript has been checked and revised.

Reviewer #2: Comments for evaluating the following manuscript

A two-gene marker for the two-tiered 1 innate immune response in COVID-19 patients

The authors addressed the two-tiered innate immune response has not been examined in COVID-19.

I would like to thank the author for revision, but I have some comments listed below.

• The goal must be written at the end of the introduction section and the authors must arrange the ideas in introduction serially to introduce the problem and how to solve. Moreover, the introduction must not have detailed materials and methods.

Thanks! The last paragraph of the Introduction section has been revised.

• The author should illustrate the role of geographical distribution on his results.

The role of geographic distribution has been added to the Discussion.

• The discussion needs to be improved regarding the causes for the similarities or differences for majority of the items.

The Discussion section has been further revised.

• Under all figures, there were many abbreviations those must be mentioned as footnotes.

The figure legends have been further revised for clarity.

---

## [Editor Report · Decision Letter 2]

28 Dec 2022

A two-gene marker for the two-tiered innate immune response in COVID-19 patients

PONE-D-22-29608R2

Dear Dr. Lei,

We’re pleased to inform you that your manuscript has been judged scientifically suitable for publication and will be formally accepted for publication once it meets all outstanding technical requirements.

Kind regards,

Benjamin M. Liu, MBBS, PhD, D(ABMM), MB(ASCP)

Academic Editor

PLOS ONE

Additional Editor Comments (optional):

ref46 needs to be corrected
---

## [Editor Report · Acceptance letter]

5 Jan 2023

PONE-D-22-29608R2 

A two-gene marker for the two-tiered innate immune response in COVID-19 patients 

Dear Dr. Lei:

I'm pleased to inform you that your manuscript has been deemed suitable for publication in PLOS ONE. Congratulations! Your manuscript is now with our production department. 

Kind regards, 

on behalf of

Dr. Benjamin M. Liu 

Academic Editor

PLOS ONE